# Muscle vs. Fasciocutaneous Microvascular Free Flaps for Lower Limb Reconstruction: A Meta-Analysis of Comparative Studies

**DOI:** 10.3390/jcm11061557

**Published:** 2022-03-11

**Authors:** Vladimir Mégevand, Domizio Suva, Morad Mohamad, Didier Hannouche, Daniel F. Kalbermatten, Carlo M. Oranges

**Affiliations:** 1Department of Plastic, Reconstructive and Aesthetic Surgery, Geneva University Hospitals, University of Geneva, 1205 Geneva, Switzerland; vladimir.megevand@gmail.com (V.M.); daniel.kalbermatten@hcuge.ch (D.F.K.); 2Department of Orthopedic Surgery, Geneva University Hospitals, University of Geneva, 1205 Geneva, Switzerland; domizio.suva@hcuge.ch (D.S.); morad.mohamad@hcuge.ch (M.M.); didier.hannouche@hcuge.ch (D.H.)

**Keywords:** muscle flap, fasciocutaneous flap, free flap, lower limb reconstruction, lower extremity microvascular reconstruction

## Abstract

(1) Background: Lower extremity microvascular reconstruction aims at restoring function and preventing infection while ensuring optimal cosmetic outcomes. Muscle (M) or fasciocutaneous (FC) free flaps are alternatively used to treat similar conditions. However, it is unclear whether one option might be considered superior in terms of clinical outcomes. We performed a meta-analysis of studies comparing M and FC flaps to evaluate this issue. (2) Methods: The PRISMA guidelines were followed to perform a systematic search of the English literature. We included all articles comparing M and FC flap reconstructions for lower limb soft tissue defects following trauma, infection, or tumor resection. We considered flap loss, postoperative infection, and donor site morbidity as primary outcomes. Secondary outcomes included minor recipient site complications and the need for revision surgery. (3) Results: A total of 10 articles involving 1340 patients receiving 1346 flaps were retrieved, corresponding to 782 M flaps and 564 FC flaps. The sizes of the studies ranged from 39 to 518 patients. We observed statistically significant differences (*p* < 0.05) in terms of donor site morbidity and total flap loss with better outcomes for FC free flaps. Moreover, the majority of authors preferred FC flaps because of the greater aesthetic satisfaction and lesser rates of postoperative infection. (4) Conclusion: Our data suggest that both M and FC free flaps are safe and effective options for lower limb reconstruction following trauma, infection, or tumor resection, although FC flaps tend to provide stronger clinical benefits. Further research should include larger randomized studies to confirm these data.

## 1. Introduction

Surgical management of complex soft tissue defects of the lower extremity has markedly improved in the last three decades through the application of free tissue transfer, substantially reducing the need for amputations [1]. While the use of skin grafts or local flaps is limited by the paucity of soft tissue in the lower extremity, advanced microvascular techniques are often preferred as they allow one to tailor free flaps to the specific defect, whether following trauma, infection, or tumor resection, providing excellent success rates in terms of definitive coverage [1].

As the field of microsurgery continues to grow, the number of available free flaps has increased over the last years, with over one hundred potential donor sites reported in the literature [1]. Two foremost surgical approaches are alternatively used and have been widely described to assess complex lower limb soft tissue defects: muscle (M) free flaps and fasciocutaneous (FC) free flaps. Both methods have been proven to be safe and effective regarding limb salvage and functional recovery.

Historically, M flaps have been preferred because of their ability to obliterate dead space and to reduce the risk of infection by providing efficient blood supply, especially in open contaminated wounds, promoting fracture healing and decreasing the incidence of infection [2,3]. Latissimus Dorsi, Gracilis, Serratus Anterior, and Rectus Abdominis muscle flaps are most commonly used according to the literature [4].

However, the use of FC flaps has become increasingly popular and favored by various authors because of the reported positive results, especially in the case of reconstruction of distal lower limb open fractures [5,6]. Free FC flaps provide thin, supple, and cosmetic soft tissue coverage, with minor donor site morbidity. The Anterolateral Thigh (ALT) flap is the most used in lower extremity microvascular reconstruction [4,5].

Although several retrospective studies have reported specific outcomes resulting from M or FC microvascular free flaps in a comparative view, there are no sustained results on the overall overperformance of one flap typology in terms of clinical outcomes. Given the lack of comprehensive data on comparative studies that evaluate this issue, we performed a meta-analysis to critically evaluate the spectrum of reported outcomes associated with M versus FC free flap coverage in lower extremity microvascular reconstruction.

## 2. Materials and Methods

The study was conducted in accordance with the Preferred Reporting Items for Systematic Reviews and Meta-Analyses (PRISMA).

### 2.1. Literature Search Methodology

An exhaustive English-language literature search was performed in May 2021 through PubMed, Cochrane Library, and ResearchGate to identify all studies on lower limb reconstruction comparing outcomes of M and FC free flaps for wound coverage. The keywords muscle flap, fasciocutaneous flap, free flap, lower limb reconstruction, and lower extremity microvascular reconstruction were used as search strings.

### 2.2. Selection Process

The titles and abstracts were independently scrutinized by two reviewers (VM and CMO) to identify relevant articles for this review according to the validated methods of the Preferred Reporting Items for Systematic Reviews and Meta-Analyses (PRISMA) statement. Any disagreement between reviewers was resolved by consensus after a consultation with a third independent reviewer (DFK). Full-text articles with patients receiving free M or FC flaps for lower limb reconstruction following trauma, infections, and tumor resection were included. Case reports, non-English articles, reviews, isolated abstracts, animal studies, and non-lower limb studies, as well as those analyzing techniques other than flap-based reconstruction, were excluded. No limitations were applied on the age of the patients, their ethnicity, or the subtype of flap used for each reconstruction. In addition, the reference lists of all relevant articles were scrutinized to identify additional relevant studies.

### 2.3. Data Extraction

After reviewing each publication, data from eligible studies were independently extracted by two authors (VM and CMO) using a standardized Excel file. The following data were collected: first author, publication year, study design, the total number of patients, total number and type of flaps performed, mean age of patients, and mean follow-up. Postoperative clinical outcomes, including infection rates, flap loss, recipient site complications, donor site morbidities, and need for revision surgery, were then assessed and compared for each type of flap. No attempt to retrieve missing data from the authors of the included papers was made.

### 2.4. Outcome Assessment

All outcomes obtained from the selected studies were reported with the same measurements retrieved from the articles and compared if homogeneous. Objective clinical outcomes were categorized as primary or secondary outcomes. Flap loss, postoperative infection, and donor site morbidity were considered as primary outcomes. Secondary outcomes included recipient site complications without implying flap loss and the need for revision surgery.

### 2.5. Statistical Analysis

Mantel–Haenszel’s method was used to combine risk ratios across studies [7]. No continuity correction was applied for studies without an event in one arm. The level of heterogeneity was measured by the I2 statistic (0% to 40%: might not be important, 30% to 60%: may represent moderate heterogeneity, 50% to 90%: may represent substantial heterogeneity, 75% to 100%: considerable heterogeneity) and tested with Cochran’s Q test [8]. If the I2 statistic was higher than 40%, a model with random effects was used with DerSominian and Laird’s approach [9]. In addition, Leave-one-out sensitivity analyses were conducted to identify influential studies. Similar methods were used to combine risk difference across studies. Statistical analyses were carried out with software R version 4.0.2 (R Foundation for Statistical Computing, Vienna, Austria) [10] and the package meta version 4.18-2 [11].

## 3. Results

The literature search yielded ten relevant articles [4,5,6,12,13,14,15,16,17,18] and sources of information on outcomes following M free flaps and FC free flaps for lower limb reconstruction (Table 1). The literature search flowchart is shown in Figure 1.

All studies were comparative studies of retrospective nature. The studies included 1340 patients receiving 1346 flaps, corresponding to 782 (57%) M free flaps, and 564 (43%) FC free flaps. The size of the study population ranged from 39 to 518 patients and the age of the patients ranged from 34 to 52 years, with a follow-up between 12 and 48 months later. The most commonly harvested M free flaps were Latissimus Dorsi, Gracilis, Rectus Abdominis, and Vastus Lateralis, while FC free flaps were mostly represented by anterolateral thigh, radial forearm, and lateral arm flaps. Four studies [5,12,13,17] involving a total of 376 patients reported results on donor site morbidity following M or FC free flaps, and pooled analysis showed a significant difference between the two treatment groups (RR 2.55, 95% CI 1.61–4.04, *p* < 0.01; Figure 2) with M flaps showing higher rates of donor site morbidity. We also observed a significant difference between the two treatment groups concerning total flap loss (RR 1.76, 95% CI 1.04–3.00, *p* = 0.04; Figure 3) favoring the FC group. These data were reported by seven studies [4,5,13,14,15,17,18]. We did not find a significant difference between the two treatment groups with regards to the rates of postoperative infection (RR 1.15, 95% CI 0.73–1.82, *p* = 0.54; Figure 4), partial flap loss (RR 1.84, 95% CI 0.93–3.64, *p* = 0.08; Figure 5), recipient site complication without implying flap loss (RR 1.24, 95% CI 0.77–2.02, *p* = 0.38; Figure 6), or need for revision surgery (RR 1.00, 95% CI 0.29–3.43, *p* = 0.99; Figure 7).

Although not statistically significant, these pooled results showed a trend of higher complications observed in the M flap group. Finally, the exclusion of most studies from the analysis using the leave-one-out sensitivity analysis did not materially change the summary estimates (Table 2).

## 4. Discussion

The present meta-analysis collects and analyzes all existing evidence on postoperative outcomes following M versus FC free tissue transfer in lower extremity reconstruction. It is the first pooled analysis on this emerging topic, showing overall quantitative outcomes. Our findings show that M and FC free flaps are similarly effective in restoring lower limb function after trauma, infection, or oncological resection. However, M flaps tend to present significantly higher rates of donor site morbidity and total flap loss compared to FC flaps.

Nowadays, free tissue transfer using microvascular surgical techniques is routine for the salvage of lower extremities and represents the state-of-the-art reconstruction after orthopedic trauma or extensive tumor resection, allowing plastic surgeons to approach more challenging cases of soft tissue defects with improved outcomes [19].

Patient and flap selection are critical steps in the setting of lower limb reconstruction. The classic indication for free flap lower extremity reconstruction is an extensive wound in the distal third of the leg or any complex wound in the upper or middle third of the leg with composite tissue loss. Major vascular or nerve injury in the lower extremity, as well as a significant loss of a muscle compartment and a large composite tissue loss beyond possibility of soft tissue or bone reconstruction are the main contraindications to the surgery [20].

Although free flap reconstruction is commonly performed within 7 to 10 days after initial consultation, recent evidence suggests that this ideal window can safely be extended to three weeks, particularly with the use of negative pressure wound therapy [21,22]. In order to suggest a relationship between the timing of free flap transfer and the final outcomes, including postoperative infections, failure rate, and bony union, Godina et al. [23] divided 532 free flaps into four groups according to the time of surgery and demonstrated that early coverage offered better results. Haykal et al. performed a meta-analysis of timing for microsurgical free flap reconstruction for lower extremity injury and suggested that early free flap reconstructions have significantly lower rates of flap loss and infection compared to late reconstruction [24].

Nowadays, the diverse complexity of lower limb defects enhances a bespoke approach with tailored flaps for each individual. Additionally, various authors agree that one should avoid the concept of the one-size-fits-all approach [25,26]. Both the M free flap and the FC free flap are excellent options for lower extremity reconstruction in patients presenting soft tissue defects following trauma, infection, or oncological resection. Furthermore, there are some specificities to each flap that have been reported in the literature. M free flaps are generally straightforward to raise and good at obliterating dead space given their bulkiness. There is a belief that they present a vascular advantage in the wound bed compared to FC free flaps; therefore, they would better promote bone healing while inhibiting bacterial proliferation [27,28,29,30,31,32,33]. However, they are more challenging to monitor postoperatively, secondary flap refinements are more difficult to assess, and they necessitate a skin graft [2,4]. On the other hand, FC free flaps, although technically more challenging to raise, do not require muscle loss from elsewhere and, once placed, present a greater similarity to the contralateral side over a skin-grafted muscle flap [2,34,35]. Furthermore, FC free tissue transfer is considered to be better suited than M free flaps for shallow defects on the distal third of the leg and ankle in the absence of massive bone or soft tissue defects [30].

All free flaps require a good arterial bed. Evidence from experimental animal models indicates that healthy and vascularized soft tissue can contribute to fracture repair in complex lower limb injuries by providing blood supply, growth factors, and mesenchymal stem cells important for osteogenesis and bone remodeling [2,36,37]. This is a good indicator for clearance of infection, as no bone can heal in the setting of persistent osteomyelitis. Several in vitro investigations have demonstrated superior vascularity in M flaps allowing bacterial clearance, antibiotic delivery, and better bone healing. Other recent clinical studies, however, suggest equivalent effects from FC free flaps in the appropriate patient with no significant difference in bony union when comparing both flaps [4,5]. Harry et al. used a rat open tibial fracture model over a 28-day period [38]. Their results have demonstrated that FC tissue had a higher vascular density compared with muscle in contact with the fracture site at all time points (*p* < 0.0001) despite accelerated healing of fractures covered by muscle. Therefore, the more advanced healing of fractures covered by muscle is not necessarily related to the vascularity of the tissues. In essence, other factors may be important in specifically promoting fracture healing, and further research on this topic should be performed. Mehta et al. [18] have used primary radiographs to compare bone healing under M and FC flaps in open tibia fractures. The aim of their study was to determine whether there was a difference in the progression of bone healing between the two types of flaps using a standardized grading system for radiographic evaluation of tibia fractures called RUST (Radiographic Union Score for Tibia). Their results demonstrate that patients with M flaps present both higher union rates and RUST scores at 6 months compared with those with FC flaps. This may suggest that M flaps, which have long been viewed as providing a more robust environment for bone healing, promote early fracture healing.

To date, little is known about the ability of flaps to deal with infection. For a long time, M flaps were recommended for the coverage of infected wounds given their ability to obliterate dead space and, as previously discussed, their supposed superior vascularity compared to the FC free flaps. Recent studies show that there is little to no significant difference in wound and fracture healing or in the clearance of infection after M free flap versus FC free flap reconstruction. However, there is a trend toward lower infection rates following FC flaps [6,39,40,41]. This is consistent with our findings. We observed similar rates of postoperative infection with no statistically significant difference (*p* = 0.54) between the two treatment groups when analyzing the results of six of the included studies [4,5,6,12,13,18], although FC free flaps seem to provide greater resistance to infection. According to Li’s study [42], preoperative wound bed inflammation is a statistically significant risk factor for postoperative wound infection (*p* < 0.001). A study conducted by Godina [23] suggests that the treatment of complex lower limb traumas either in the acute period (<72 h) or in the chronic period following multiple debridements (>90 days) reduces the risk of postoperative infection, strongly suggesting the incidence of preoperative recipient site inflammation on postoperative wound infection. Additionally, an important factor in lower extremity reconstruction is ensuring suitability of the wound prior to soft tissue coverage. Therefore, it is often suggested that multiple thorough debridements be performed before flap transfer to increase the postoperative flap survival rate and decrease the incidence of infection. Culture-directed antibiotic therapy remains a key step to assess the clearance of infection.

Occasionally, free flaps fail. Our findings suggest that there are significantly more flap losses following M free flap transfers. Eight studies [4,5,12,13,14,15,17,18] included in our meta-analysis evaluated the incidence of flap loss following lower limb reconstruction. Our pooled analysis showed a statistically significant increase (*p* = 0.04) in total flap loss with M flaps, although the rates of partial loss were similar between the two treatment groups (*p* = 0.08). The etiologic factors associated with failure of free tissue transfer are multifactorial. Khouri [43] has classified them into three categories: preoperative factors such as patient age and comorbidities, intraoperative factors including choice of donor site and of recipient vessel, and finally postoperative factors such as flap care and anticoagulant protocol administration.

The morbidity of muscle and perforator-based flaps has been compared extensively in the literature regarding breasts, but the field of lower extremity reconstruction is still to be explored. Regarding complications not implying flap loss such as the development of a hematoma or a seroma on the recipient site, our pooled analysis showed similar results (*p* = 0.38) between the two treatment groups when combining the results of five studies [12,14,15,16,17], although M free flaps tend to present slightly higher rates of recipient site complications in comparison with FC free flaps.

Traditionally, M flaps are known for engendering higher rates of donor site morbidity than FC flaps; however, there is evidence suggesting that the results are similar between the two techniques [44]. Interestingly, in the setting of debilitated and injured trauma patients, the preservation of core muscle units such as the Latissimus Dorsi and the Rectus Abdominis which are necessary for rehabilitation, offers better functional outcomes and therefore makes FC flaps preferable for some authors [45]. Our findings show significantly higher rates of donor site morbidity when using M flaps (*p* < 0.01) rather than FC flaps after analyzing the results of four studies [5,12,13,17] included in our meta-analysis. All authors agree that there is less functional impairment following FC free tissue transfer as these flaps imply a less extensive tissue harvesting.

An important factor that should be discussed with patients prior to surgery is the aesthetic outcome of the various flap types. Patients’ perception of the success of their surgery partly resides in the final look of their reconstructed limb and its aesthetic appearance. M flaps typically appear bulky at first and later experience significant atrophy and improved contouring [12]. FC flaps on the other hand do not undergo these changes and can appear quite bulky over time. Seyidova [12] demonstrated a significant difference (*p* = 0.003) in patient satisfaction regarding flap texture after surgery in favor of FC free flaps when compared with skin-grafted muscle flaps, although the contour and color match did not significantly differ. Overall, aesthetic satisfaction both on the donor site (*p* = 0.002) and the recipient site (*p* = 0.001) was significantly higher after FC flaps in Philandrianos’ [17] 2018 retrospective study on 47 patients. Finally, Cho et al. [14] in their multi-center study for lower limb reconstruction using free flaps estimated that almost 30% of their patients had undergone aesthetic refinement procedures, although no differences were noted in the rates of secondary flap refinement for cosmetic purposes between M and FC free flaps. Aesthetic consideration after free flap transfer should especially be a concern in younger patients and should be considered an integral part of lower extremity reconstruction.

Four studies [4,13,15,17] included in our meta-analysis reviewed the need for revision surgery other than for cosmetic purposes following M and FC free tissue transfer. These procedures imply debridement, split-thickness skin grafting, flap debulking, or extremity amputation, depending on the severity of the complication. Our pooled analysis showed almost identical results between the two treatment groups (*p* = 0.99). However, evidence from previous reports seems to be inconclusive and conflicting on this topic as most of these studies are small case series and not thorough enough to establish statistical significance.

The results of this meta-analysis should be viewed with caution due to the number of limitations and potential biases influencing these findings. This meta-analysis collects information extracted from comparative studies evaluating the benefits of M versus FC free flaps for lower limb reconstruction. Although most of the included studies [5,6,13,15,17,18] selected patients who underwent surgery following traumatic injuries only, several others [4,12,14,16] combined trauma, chronic infection, and oncological excision as the main indications for reconstruction. We therefore collected all of the information provided in the selected studies and compared the results all together. This is a major limitation to our findings as all these wounds in clinical routine are very different in terms of microbiological contamination, vascularization, involved tissue components, comorbidities, and time of the wound. Moreover, all studies used for this pooled analysis were of retrospective nature. However, there was a strong homogeneity in methods and settings for two relevant outcomes, total flap loss and partial flap loss, with, respectively, 0% and 6% heterogeneity between the studies. To minimize heterogeneity and reduce bias, we only included comparative studies, excluding outcomes of one-arm studies. Other major limitations of this analysis include the lack of randomization in the included studies, leading to a selection bias as there is no way to determine the intraoperative decision-making process that guided flap selection for each case. As reported in the literature, most surgeons tend to use muscle flaps in the setting of complex trauma as they offer greater dead space obliteration. By definition, these patients are inevitably at greater risk of undergoing secondary procedures leading to higher rates of postoperative complications.

## 5. Conclusions

Our data suggest that M and FC free flaps are both safe and effective approaches to lower extremity microvascular reconstruction after trauma, infection, or oncological excision, although FC free flaps offer significantly less donor site morbidity and seem to provide both greater aesthetic satisfaction and better environment-reducing susceptibility to postoperative infection. However, the results of this meta-analysis should be interpreted with caution because it only includes retrospective studies, and therefore larger randomized investigations should be performed to verify these data. Both reconstructive options must be included in the armamentarium of the plastic surgeon who must be able to select the best option according to individual needs.

## Figures and Tables

**Figure 1 jcm-11-01557-f001:**
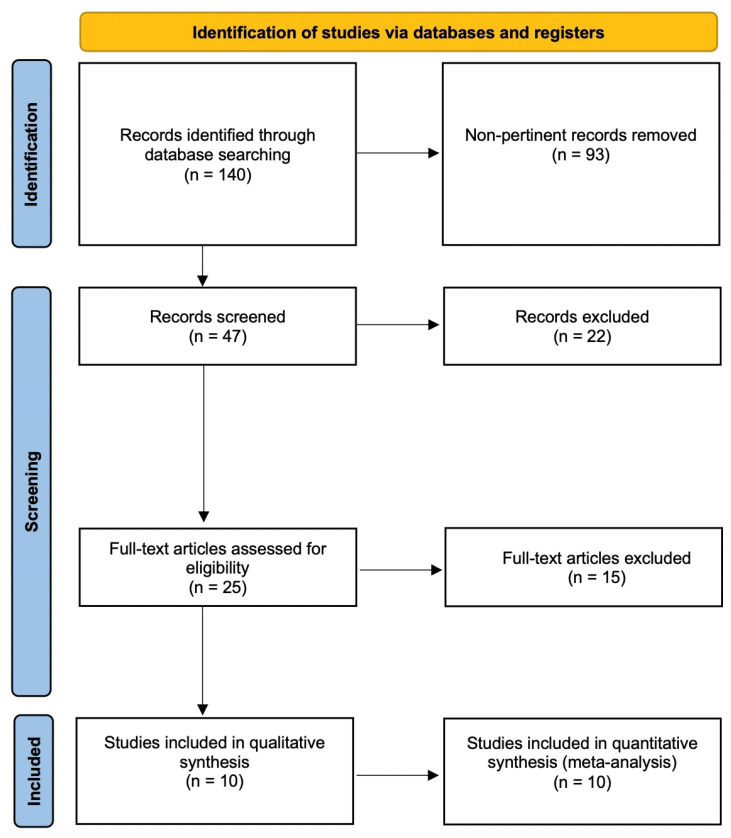
Literature search flowchart.

**Figure 2 jcm-11-01557-f002:**
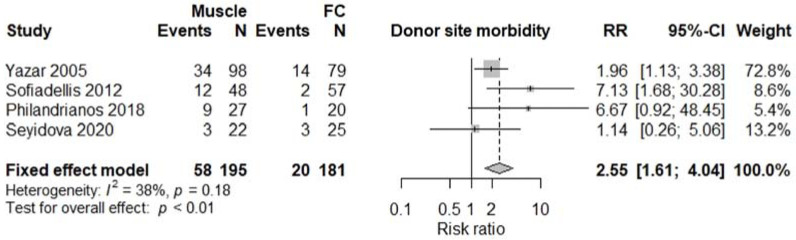
Donor site morbidity.

**Figure 3 jcm-11-01557-f003:**
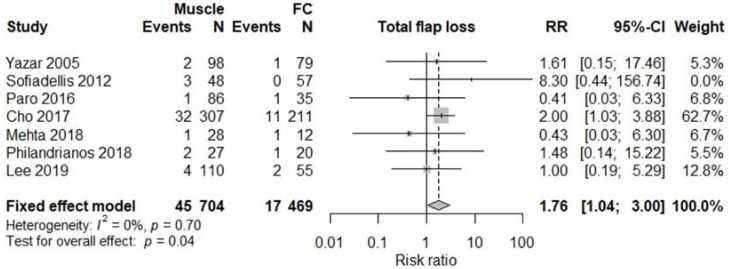
Total flap loss.

**Figure 4 jcm-11-01557-f004:**
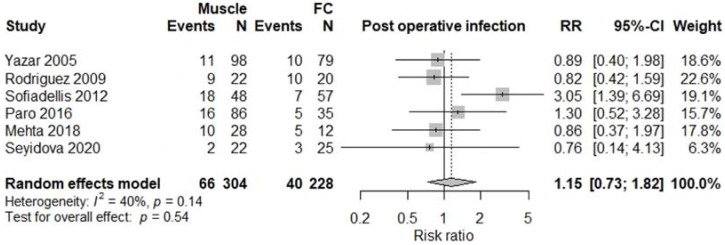
Postoperative infection.

**Figure 5 jcm-11-01557-f005:**
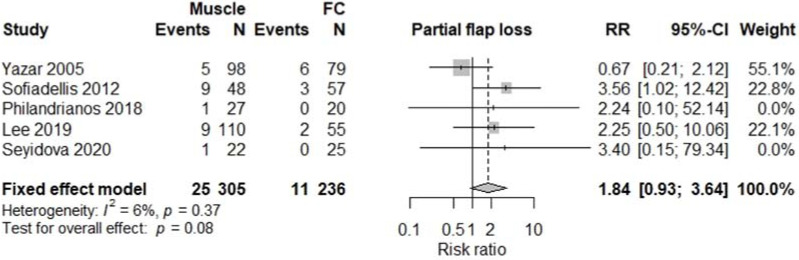
Partial flap loss.

**Figure 6 jcm-11-01557-f006:**
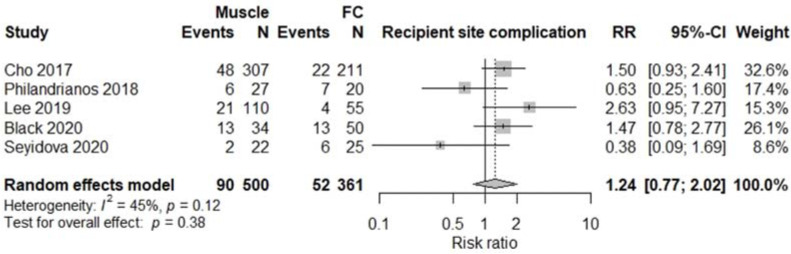
Recipient site complication not implying flap loss.

**Figure 7 jcm-11-01557-f007:**
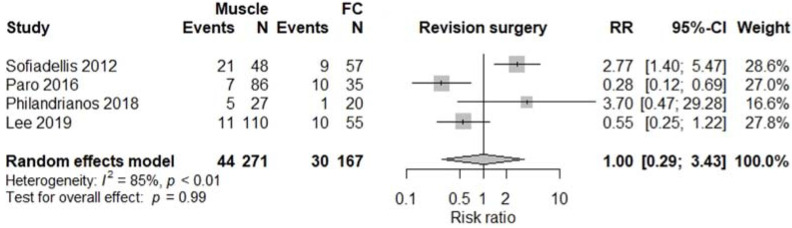
Need for revision surgery.

**Table 1 jcm-11-01557-t001:** Characteristics of the included studies.

Author	Patients	Flaps	M Flaps	FC Flaps	Mean Age M (Years)	Mean Age FC (Years)	Follow-Up (Months)
Yazar 2005	174	177	98	79	34.6	36.3	24
Rodriguez 2009	42	42	22	20	43	40	32.5
Sofiadellis 2012	103	105	48	57	42.5	41.3	12
Paro 2016	121	121	86	35	46.9	49.9	-
Cho 2017	518	518	307	211	-	-	48
Mehta 2018	39	40	28	12	37	33.5	12
Philandrianos 2018	47	47	27	20	36.1	40.1	34.2
Lee 2019	165	165	110	55	35	35	-
Black 2020	84	84	34	50	52.8	58.5	11
Seyidova 2020	47	47	22	25	48	50	-

**Table 2 jcm-11-01557-t002:** Leave-one-out sensitivity analysis.

		Pooled RR			Pooled RD		
Outcome	Removed Study	Estimate (95% CI)	*p*	*I*^2^ (%)	Estimate (95% CI)	*p*	*I*^2^ (%)
Postop infection	Yazar 2005	1.35 (0.92 to 1.96)	0.1227	49	0.072 (−0.017 to 0.160)	0.1116	50
	Rodriguez 2009	1.36 (0.92 to 2.02)	0.1268	41	0.054 (−0.014 to 0.121)	0.1175	56
	Sofiadellis 2012	0.93 (0.63 to 1.38)	0.7176	0	−0.014 (−0.086 to 0.059)	0.7148	0
	Paro 2016	1.22 (0.85 to 1.76)	0.2875	51	0.041 (−0.034 to 0.117)	0.2821	58
	Mehta 2018	1.30 (0.90 to 1.89)	0.1654	48	0.049 (−0.018 to 0.116)	0.1542	57
	Seyidova 2020	1.26 (0.89 to 1.79)	0.1871	51	0.049 (−0.022 to 0.120)	0.1778	56
Donor site morbidity	Yazar 2005	4.13 (1.73 to 9.85)	0.0014	45	0.184 (0.088 to 0.279)	0.0002	51
	Sofiadellis 2012	2.12 (1.29 to 3.47)	0.0029	1	0.162 (0.067 to 0.258)	0.0009	45
	Philandrianos 2018	2.31 (1.44 to 3.71)	0.0005	44	0.162 (0.078 to 0.247)	0.0002	30
	Seyidova 2020	2.76 (1.69 to 4.50)	<0.0001	49	0.200 (0.115 to 0.285)	<0.0001	0
Total flap loss	Yazar 2005	1.77 (1.03 to 3.05)	0.0392	0	0.033 (0.003 to 0.062)	0.0302	16
	Sofiadellis 2012	1.61 (0.94 to 2.76)	0.0854	0	0.025 (−0.002 to 0.053)	0.0703	9
	Paro 2016	1.86 (1.08 to 3.22)	0.0254	0	0.033 (0.006 to 0.061)	0.0176	1
	Cho 2017	1.37 (0.56 to 3.34)	0.4918	0	0.009 (−0.018 to 0.037)	0.5055	0
	Mehta 2018	1.86 (1.08 to 3.21)	0.0258	0	0.033 (0.005 to 0.057)	0.0181	23
	Philandrianos 2018	1.78 (1.03 to 3.07)	0.0380	0	0.025 (0.003 to 0.055)	0.0291	28
	Lee 2019	1.88 (1.07 to 3.29)	0.0282	0	0.033 (0.005 to 0.061)	0.0203	24
Partial flap loss	Yazar 2005	3.27 (1.27 to 8.40)	0.0139	0	0.071 (0.021 to 0.122)	0.0059	0
	Sofiadellis 2012	1.33 (0.57 to 3.10)	0.5098	0	0.015 (−0.027 to 0.057)	0.4868	0
	Philandrianos 2018	1.77 (0.89 to 3.51)	0.1037	29	0.039 (−0.006 to 0.084)	0.0901	43
	Lee 2019	1.72 (0.80 to 3.69)	0.1641	27	0.036 (−0.015 to 0.087)	0.1617	42
	Seyidova 2020	1.75 (0.88 to 3.48)	0.1116	26	0.038 (−0.007 to 0.083)	0.0956	43
Recipient site complic.	Cho 2017	1.10 (0.53 to 2.26)	0.7997	56	0.013 (−0.134 to 0.160)	0.8639	61
	Philandrianos 2018	1.45 (0.91 to 2.31)	0.1162	32	0.056 (−0.026 to 0.139)	0.1827	47
	Lee 2019	1.10 (0.66 to 1.83)	0.7205	44	0.002 (−0.111 to 0.115)	0.9710	46
	Black 2020	1.12 (0.56 to 2.25)	0.7432	58	0.022 (−0.077 to 0.121)	0.6661	59
	Seyidova 2020	1.40 (0.91 to 2.15)	0.1260	32	0.067 (0.006 to 0.129)	0.0328	19
Revision surgery	Sofiadellis 2012	0.59 (0.21 to 1.67)	0.3211	63	−0.055 (−0.231 to 0.121)	0.5423	76
	Paro 2016	1.57 (0.43 to 5.75)	0.4919	80	0.104 (−0.121 to 0.329)	0.3642	85
	Philandrianos 2018	0.77 (0.20 to 3.05)	0.7140	89	−0.005 (−0.263 to 0.253)	0.9688	89
	Lee 2019	1.31 (0.22 to 7.61)	0.7673	88	0.069 (−0.219 to 0.357)	0.6391	89

## Data Availability

Not applicable.

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
