# Peer review of "Muscle vs. Fasciocutaneous Microvascular Free Flaps for Lower Limb Reconstruction: A Meta-Analysis of Comparative Studies"

_jcm, 2022, doi:10.3390/jcm11061557_

Round 1

Reviewer 1 Report

Dear Editor, Dear authors,

Thank you very much giving us the opportunity to review the manuscript entitled " Muscle vs Fasciocutaneous microvascular free flaps for lower limb reconstruction: a meta-analysis of comparative studies".

This very interesting article describes the meta-analysis of studies comparing M and FC flaps.

The manuscript is written well. Tables are presenting the results. Figures / illustrations are missing. The findings are discussed uncritical.

From our point of view, there is one major points which need to be addressed.

The authors wrote “On the basis of all available evidence, this comprehensive meta-analysis of the literature demonstrates that FC free flaps offer significantly less donor site morbidity and provide a better environment, reducing susceptibility to postoperative infection in case of lower extremity microvascular reconstruction after trauma, infection, or oncological excision.”

The problem is that they have analyzed all indication “trauma, infection, or oncological excision” together, but that wounds in clinical routine are very different concerning microbiological contamination, vascularization, involved tissue components, comorbidities and time of the wound. This are key points triggering the indication FC versus M flap. So only including these aspect and a differential analyses allows us to draw any conclusions for clinical routine and will help us in decisions making.

The manuscript must be improved changing these aspects according to our recommendations.

Author Response

Response to Reviewer 1 Comments

Point 1: From our point of view, there is one major points which need to be addressed.
The authors wrote “On the basis of all available evidence, this comprehensive meta-analysis of the literature demonstrates that FC free flaps offer significantly less donor site morbidity and provide a better environment, reducing susceptibility to postoperative infection in case of lower extremity microvascular reconstruction after trauma, infection, or oncological excision.” The problem is that they have analyzed all indication “trauma, infection, or oncological excision” together, but that wounds in clinical routine are very different concerning microbiological contamination, vascularization, involved tissue components, comorbidities and time of the wound. This are key points triggering the indication FC versus M flap. So only including these aspect and a differential analyses allows us to draw any conclusions for clinical routine and will help us in decisions making.
The manuscript must be improved changing these aspects according to our recommendations.

ANSWER: Thank you very much for your comments and suggestions to our paper. We agree with your observations. Although most of the included papers focused their research on one same indication for reconstruction, namely severe trauma, others recruited patients presenting with different etiologies to their soft tissue defect, namely trauma, chronic infection and oncological excision. This meta-analysis aims at collecting all available data regarding muscle and fasciocutaneous microvascular free flaps for lower limb reconstruction in a comparative view. We therefore extracted the results provided by all existing comparative studies on this topic and aimed at providing a global comprehensive analysis on the matter, reflecting the current knowledge. The retrospective nature of these studies provided a selection bias in the type of flaps used for each indication, which represents a major limitation to our findings. We added a paragraph to the Discussion section to highlight this aspect and add your comments.

Reviewer 2 Report

I commend the authors for organizing and performing this systematic review and meta analysis study which compares the free muscle and fasciocutaneous flaps on the reconstruction of the leg and ankle. I believe that this study will be a helpful tool for the reader when seeking evidence for the pros and cons of each free flap option on the lower extremity reconstruction.

Author Response

Dear Reviewer,
We thank you very much for you kind comments and for your appreciation of our article.